# External Field-Controlled Ablation: Magnetic Field

**DOI:** 10.3390/nano9121662

**Published:** 2019-11-22

**Authors:** Jovan Maksimovic, Soon Hock Ng, Tomas Katkus, Bruce C. C. Cowie, Saulius Juodkazis

**Affiliations:** 1Centre for Micro-Photonics, Faculty of Science, Engineering and Technology, Swinburne University of Technology, John Street., Hawthorn, VIC 3122, Australia; soonhockng@swin.edu.au (S.H.N.); tkatkus@swin.edu.au (T.K.); 2Australian Synchrotron, 800 Blackburn Road, Clayton, VIC 3168, Australia; brucec@ansto.gov.au; 3Melbourne Center for Nanofabrication, Australian National Fabrication Facility, Clayton, VIC 3168, Australia; 4Tokyo Tech World Research Hub Initiative (WRHI), School of Materials and Chemical Technology, Tokyo Institute of Technology, 2-12-1, Ookayama, Meguro-ku, Tokyo 152-8550, Japan

**Keywords:** ablation, magnetic field, femtosecond laser fabrication

## Abstract

The femtosecond laser ablation of silicon amidst an externally applied magnetic field in different orientations was investigated with respect to the scanning direction and polarisation of the laser beam, by observation of ablation patterns and debris displacement in a range of fluences, magnetic fields strengths, and geometries. Ultra-short ∼230 fs laser pulses of 1030 nm wavelengths were utilised in the single and multi-pulse irradiation modes. Ablation with an externally applied magnetic B-field $B_{ext}\approx 0.15$ T was shown to strongly affect debris formation and deposition. The mechanism of surface plasmon polariton (SPP) wave can explain the ablated periodic patterns observed with alignment along the magnetic field lines. The application potential of external field controlled ablation is discussed.

## 1. Introduction

Ultra-short laser pulses provide nanoscale resolution in 3D polymerisation, optical waveguide inscription in glasses and crystals [1], non-erasable optical memory and photonic crystals [2,3,4], nano-structuring of surfaces and in the bulk [5,6,7,8], creation of new materials and their high pressure/temperature phases by 3D confined micro-explosions [9,10,11], thermal morphing of laser fabricated 3D structures [12], and laser assisted etching [13,14,15]. Applications of colloidal nanoparticle synthesis by ablation in liquids [16] and laser machining have become industrial applications with a high throughput [17,18,19,20]. In all the aforementioned studies, morphological changes on the surface or inside the volume of an irradiated sample are exploited for different applications; e.g., formation of nanogratings for optical retardance, wettability change of the surface, alteration of wet etchability, change of ablation debris formation, and change of ablation rate. The possibility to influence the light-matter interaction inside the laser affected focal volume by applying an external electric and magnetic field is a promising line of research for the understanding of fundamental aspects of laser breakdown and solid state plasma. In the case of long nanosecond laser pulses when strong plasma shielding exists in front of the irradiated surface, some acceleration of ablation rates were observed for stainless steel and silicon [21]. Plasma-assisted surface processing of Si at low NA focusing conditions using 0.3 ps pulses with ethanol as the medium and in the presence of an externally applied repulsive magnetic field showed the benefit of producing smoother surfaces; however, the ablation volume was smaller [22]. Both of those recent studies used externally applied fields fabricated with laser pulses that heavily overlapped with larger focal volumes than that used in this study. This calls for further studies for ablation in an externally applied B-field under unexplored tight focusing conditions with focal spot sizes of a few wavelengths and when the breakdown plasma screening is not affecting energy delivery by fs-laser pulses. For this, an exact intensity at the focal spot and B-field values should be directly measured. One particular motivation of this study was to observe a B-field effect on debris formation, which could possibly be utilised for spatial sorting of high-refractive index nanoparticles. Small tens-of-nm nanoparticles of Si have promising applications in optical and bio-medical sensing due to support of electric and magnetic dipole modes in light scattering [23].

Here, we introduce a new control method over the ablation process using an externally applied magnetic B-field. With a Si sample placed in a millimetre-scale gap between Nd-magnets of ∼0.1–0.2 T, the ablation plume, along with the ablated surface of crystalline Si on which a surface plasmon polariton (SPP) wave was induced were strongly affected. The orientation of magnetic B-field rather than the polarisation of the laser lights’ E-field strongly governs the formation of the imprinted patterns of aligned micro-ablation pits. The laser ablated patterns are determined by the instant electron densities in the surface layer, their velocity and magnetic field strength providing the ability to control deposition in the femtosecond laser regime. The displacement of debris was also shown to be heavily affected by a very weak B-field with very low incident fluence, inciting an application to machining and sensing applications. In a comprehensive manner this study also shows, in detail, the effect of the polarisation of light and fluence on magnetic field-induced phenomena in ultra-short pulse fabrication.

## 2. Results

### 2.1. Peculiarities of Ablation Patterns in the External *B*-Field

Ablation was carried out at high intensities, typical for laser cutting applications, exceeding the ablation threshold more than ten times. The direction of ablation plume and polarisation of the incident light E-field were both perpendicular to the magnetic field to maximise the Lorenz force F=qE+q[v×B] for charges *q* in the ablation plume. Patterns of periodic small ablation pits were formed along the applied B-field lines, extending well beyond the focal spot diameter, with the ablation site on the central axis of the magnets and central on the sample surface relative to its shape (on Si $\langle \bar{1}10\rangle$). Separation between the adjacent pits were close to the wavelength of laser pulse and a surface plasmon polarisation (SPP) was induced at a Si–air interface. Without a magnetic field present B=0, a circularly symmetric ablation pit, was formed characteristic of the measured Gaussian laser beam profile, as shown in Figure 1. Distinctions between single pulse ablation sites for the B=0 and B≠0 cases are well expressed for every pulse irradiated between the magnets or even in a close proximity of magnets.

Figure 1 shows the observed morphology change of single-pulse ablation sites on Si in the presence of an externally applied magnetic field $B_{ext}$ = 0.1–0.2 T; such magnetic fields are close to the maximum achievable strength with standard Nd-magnets. A rhomboidal structure around the central elliptically elongated ablation pit has an approximate side orientation perpendicular to the Si directions 〈010〉 and 〈100〉. The Lorenz force acting on electrons is directed in (out) to the samples surface when the linearly polarised light field is along the y-axis ($v_{y}\propto E_{y}$) while the B-field is perpendicular along the x-axis ($B_{x}$; Figure 1). In this geometry, free and bound electrons follow an oscillation normal to the Si surface following $F_{z}$. It was previously demonstrated that surface ablation in the direction of E-field polarisation, when it is perpendicular to the surface, is the most efficient in material removal [24]. Ablation pits extending beyond the focal spot of 2r=1.8μm can be explained by the SPP mechanism, as discussed below (Section 3).

Figure 2 presents a summary of the observed dependencies of the ablation pattern versus the gap size *x*, light polarisation (E-field), magnetic B-field orientation, and its strength (the gap size *x*). The most pronounced orientation dependence is a linearly aligned pattern of ablated circular pits along the B-field orientation. The period of the pattern was changing almost linearly from Λ=1.22μm for the gap x=5 mm to 2.3 μm for x=2 mm. When the gap between magnets was larger than 5 mm, the pattern of linearly aligned micro-pits disappeared and became similar to pulses without applied magnetic fields (magnets removed from the holder). The effects of a circularly polarised E-field were explored to further detail the alignment of produced patterns; however, the period and alignment were not changed (not shown here for brevity). All central regions of pulses affected by externally applied fields can also be seen to become elliptical in the direction of the magnetic field’s orientation. Figure 3 shows a detailed morphology evolution of the ablated sites for different orientations of the linearly polarised fs-laser pulses. The period $\Lambda_{s}$ of the linearly formed pattern seen in ablated micro-pits was always aligned to the magnetic B-field and was not affected by the polarisation of E-field of incident laser pulses. In Figure 3d,e it can be observed how repeatable the phenomena is, as different pulses irradiated side by side are almost completely identical under the same conditions.

Narrow Si samples were tilted inside the magnet gap to check the effect of B-field orientation relative to change of the *E*-field orientation of incident light. The pattern of linearly aligned ablation pits was always aligned along the B-field direction while the orientation of the *E*-field had no effect on the orientation of micro-pits and no coupling was observed.

The patterns of aligned ablated micro-pits with periodic separation of ∼1–2.5 μm were dependent on the spacing between magnets *x*, resembling a pattern of the SPP wave. A SPP has a wavelength close to the free space wavelength $\lambda^{spp}\approx \lambda ∼1\;\mu$m. The period of ablated pits observed depends on the gap *x*; i.e., on the B-field strength. The linear patterns of ablation pits were also observed at lower pulse energies $E_{p}\approx$ 10–20 nJ (on the sample, Figure A2); however, the most distinct features were observed at the highest pulse energies used and presented in Figure 1, Figure 2, Figure 3, Figure 5, and Figure A1. Figure 4a shows the scaling of the single-pulse ablation site when pulse energy is changed by a factor of $10^{2}$.

There was air breakdown above the Si surface at many of the utilised pulse energies which were very high, and one could expect a change of ablation pattern on the Si surfaces. However, SEM images in Figure 4a show a continuous change in diameter of the ablation sites. The molecular number density of air is $∼2.5\times 10^{19}$ cm${}^{-3}$ which causes intensity clamping during filamentation in air breakdown by ultra-short laser pulses; however, even full ionisation does not reach the critical plasma density at a 1030 nm wavelength which is $1.05\times 10^{21}$ cm${}^{-3}$. Intensity distribution at the NA=0.7 focusing employed is shown in Figure 4c for a low intensity case.

Figure 5 shows the dependence of pulse energy on the linear pattern period $\Lambda_{s}$. Even at low pulse energy (in the tens-of-nJ), the linear pattern (aligned to the B-field direction) begins to manifest physically after ablation, and as seen in SEM images, it evolves into multiple ablation micro-pits at higher energies. The $\Lambda_{s}$ was almost independent of pulse energy $E_{p}$ but follows a close-to-linear dependence on the B-field strength (*x* gap width), as shown in Figure 5b. This is further illustrated by 3-dimensional images in the Appendix. Extrapolation of this dependence shows that the threshold field to observe the linear pattern forming after ablation was B≈0.11 T with the utilised range in energy of incident laser pulses presented in this study.

### 2.2. Debris Dispersion

When Si samples were placed under a magnetic disk or washer (magnet with a central hole) for ablation, extensive debris formation was observed (Figure 6). When the velocity of the ablation plume *v* was parallel to the magnetic field *B*, there was no charge separation since the Lorentz force [v×B]=0. However, whenever charges started to move at an angle to the magnetic field lines, separation of electrons and ions existing in the laser ablated plume occurred. The charges ±|q| experienced a cyclotron spinning around the B-field lines with a frequency defined by their mass $\omega_{c}=qB/m$. The radius of spinning trajectory was $r_{c}=mv/\left(qB\right)$. Apparently, such charge separation in plasma and acquisition of an angular momentum favours longer debris travel times and distances. This favours a longer oxidation time and SiO_2_ formation—which some of the debris on the surface of Si sample was comprised of (in SEM images presented in Figure 6).

The near edge X-ray absorption fine structure (NEXAFS) technique was used to identify chemical modification of laser ablated regions. NEXAFS is a useful technique; e.g., to detect phase modifications of olivine [25] in laser irradiation conditions similar to those used in this study. NEXAFS confirmed an oxide on the surface of laser ablated patterns. Measurements were carried out at the O1s electron binding energy in SiO_2_ window around 532.9 eV (soft X-ray beamline of the Australian Synchrotron). Si was ablated with sample placed on the surface of a Nd-magnet the same way as in the experiment on debris generation (Figure 6) and pristine Si was used as a reference sample for comparison of oxides natively on surface and oxides produced during ablation (with and without externally applied magnetic fields).

The origin of the newly observed periodic SPP-like pattern of ablated micro-pits aligned to B-field lines is discussed next.

## 3. Discussion

### 3.1. Surface Plasmon Polariton

The surface plasmon polariton (SPP) wave at the Si–air interface can be launched when the energy and momentum conservation laws are satisfied. The wave vector of the SPP $k^{spp}=k_{1}^{spp}+ik_{2}^{spp}$ is given by:(1)$$k_{1}^{spp}=k_{0}\sqrt{\frac{\varepsilon_{1}^{Si}\varepsilon_{1}^{air}}{\varepsilon_{1}^{Si}+\varepsilon_{1}^{air}}},$$
(2)$$k_{2}^{spp}=k_{0}{\left(\sqrt{\frac{\varepsilon_{1}^{Si}\varepsilon_{1}^{air}}{\varepsilon_{1}^{Si}+\varepsilon_{1}^{air}}}\right)}^{3}\times \frac{\varepsilon_{2}^{Si}}{2{\left(\varepsilon_{1}^{Si}\right)}^{2}},$$
where $\varepsilon^{Si}\equiv \varepsilon_{1}^{Si}+i\varepsilon_{2}^{Si}=12.709+i1.7146\times 10^{-3}$ is the permittivity of Si at λ=1030 nm wavelength [26], $k_{0}=2\pi /\lambda$ is the wavevector in free space, and $\varepsilon^{air}\equiv \varepsilon_{1}^{air}=1$ is the permittivity of air. The SPP wavelength $2\pi /k_{1}^{spp}=1.07\;\mu$m (Equation (Equation 1)) and the propagation length is $1/(2k_{2}^{spp})=17.3$ mm (Equation (Equation 2)).

When the electron density in Si increases during laser pulse irradiation, the real part of permittivity $\varepsilon_{1}^{*Si}$ (∗ marks photo-excited state) decreases and imaginary part $\varepsilon_{2}^{*Si}$ increases due to free carrier absorption. The dielectric breakdown is defined by $\varepsilon_{1}=0$. A laser-induced air breakdown occurs at electrical field $E_{0}=3\times 10^{6}$ V/m, and at the utilised irradiation conditions, it is reached at a point above the sample surface. This influences the amount and the area of light energy deposition on the surface of Si. Only at lower pulse energies did the laser ablation pits have diameters similar to that of the focal spot 1.8μm (Figure 4a, Figure 5, and Figure A2). Numerical modelling of linear intensity with geometrical locations of the focus on the Si surface (Figure 4c) shows a higher intensity of light field $E^{2}$ above the samples surface due to interference with the back-reflected light. Due to this interference, an initiation of air breakdown is expected at high intensities.

An increase of the period between the ablated pits for smaller gaps of *x* (a stronger *B*-field) can be explained by ionisation of air and/or Si. Indeed, when $\varepsilon_{1}^{*air}=0.75$ in Equation (Equation 1), the period $\Lambda_{s}=2\pi /k_{1}^{spp}=1.22\;\mu$m, as was experimentally observed for the larger gaps between magnets x=4–5 mm (Figure 2a).

The line of ablation pits formed by the SPP had a period dependent on B-field strength and its orientation always followed the magnetic field lines $N\overset{}{\to }S$. Reduction of the $\varepsilon_{1}^{*Si}$ of photo-excited Si [27] can explain the increase of $\Lambda_{s}$ due to air breakdown which directly follows from Equation (Equation 1). Variation of $\varepsilon_{1}^{*Si}$ with respect to *B* should be caused by the different electron densities $N_{e}$, which are proportional to the absorbed energy defined by the absorbance A=1−R; *R* is the reflectivity coefficient. The absorbed energy density [J/cm${}^{3}$] at the end of the laser pulse is $W_{a}=2AF_{p}/l_{abs}$ [28], where the fluence $F_{p}=\int_{0}^{t_{p}}I_{p}\left(t\right)dt$; the absorption depth for the E-field is $l_{abs}=c/\left(\omega k\right)=2\pi /\kappa$, dependent on the refractive index $\sqrt{\varepsilon }=n+i\kappa$ and $A=4n/[{\left(n+1\right)}^{2}+\kappa^{2}]$; and the depth of absorption is $l_{abs}/2$ since $I\propto E^{2}$ [29].

The electron density $N_{e}$ at the end of the pulse is defined by $\int_{0}^{t_{p}}I_{p}\left(t\right)dt$, as described above. The average intensity of the laser pulse is $I_{a}=c\varepsilon_{0}E_{0}^{2}/2\equiv cB_{0}^{2}/\left(2\mu_{0}\right)\equiv E_{0}B_{0}/\left(2\mu_{0}\right)$, where $E_{0}$ is the amplitude of the electrical field strength [V/m], $B_{0}=E_{0}/c$ is the maximum magnetic field strength [T], $\varepsilon_{0},\mu_{0}$ are the permittivity and permeability of the free space, respectively. Generation of electrons by laser pulses, follow in time with the instantaneous intensity I∝EB, where the lights field $B_{h\nu }\approx 60$ T ($E_{p}=1.13\;\mu$J; Figure 1) while Nd_2_Fe_14_B magnets added $B_{ext}=$ 0.1–0.2 T, a negligible contribution; the hν marker utilised is for the light field. The dependence of $\Lambda_{s}$ period with *B*-field strength as well as the alignment $N\overset{}{\to }S$ of the pattern should be linked to the plasma density at the end of the pulse (defining $\varepsilon_{1}^{*Si}$) and ensuing plasma dynamics in the external B-field (after the light pulse). Due to the vectorial nature of Lorentz force, electrons are coiled around the magnetic field lines with different cyclotron frequency and rotation radius.

Polarisation of laser pulses can define the orientation of oscillation in electrons—the velocity vector $v_{e}$—the applied light field $v_{e}∥E$, was not an important factor for formation of SPP-like patterns.

The phase mismatch of the SPP wave vector $k_{1}^{spp}=5.87\times 10^{6}$ m${}^{-1}$ with that of the laser light $k_{0}=6.1\times 10^{6}$ m${}^{-1}$ is accommodated by scattering and redirection from a strongly localised scattering point, the d=1.8μm diameter focal spot, from which light is scattered into wide range of angles $\theta_{s}$. The condition $k_{1}^{spp}=k_{0}\pm \cos\theta_{s}\times \left(2\pi /d\right)$ would launch an SPP on the surface of photo-excited Si with the period defined by its permittivity and direction governed along the orientation of the external B-field; here $2\pi /d=3.5\times 10^{6}$ m${}^{-1}$. The external field was contributing to the absorbed light energy in a pre-surface skin layer of Si of $1/l_{abs}$ depth, as discussed above.

### 3.2. B-Field Generation

At high laser intensity, energy deposition into the skin (absorption) depth becomes sub-wavelength. This favours the formation of strong electron concentration and temperature gradients and can lead to spontaneous generation of magnetic fields [30]. When the electron temperature gradient $\nabla T_{e}$ and the concentration gradient $\nabla N_{e}$ are not parallel, $\partial B/\partial t=\left[\nabla T_{e}\times \nabla N_{e}\right]/\left(eN_{e}\right)$ [30]. Plasma dynamics immediately after the laser pulse are influenced by the externally applied B-field, as discussed for the debris formation; however, in Si it caused SPP-like aligned ablation micro-pits. A stronger B-field causes a larger period $\Lambda_{s}$ for the same laser pulse energy (Figure 5b). Low-conductivity Si was utilised and the photo-generated carrier dynamics starts from the optically excited volume with a lateral cross section ∼2μm, and a few micrometers of depth. Further studies are planned for determination of the mechanism of SPP-like ablation patterns.

## 4. Materials and Methods

Direct laser writing was carried out with laser fabrication integrated system (Altechna, Ltd. Vilnius, Lithuania.) comprised of $t_{p}$ = 230 fs pulse duration and λ=1030 nm wavelength pulsed laser (Pharos, Light Conversion, Ltd. Vilnius, Lithuania.) and high precision mechanical stages (Aerotech, GmbH. Pittsburgh, USA.). All laser fabrication was carried out in air (Cleanroom, Class 1000). The laser beam profile was measured utilising a beam profiling camera (Spiricon SP928, Ophir. North Logan, USA).

The magnet and sample holding jigs with variable magnet spacing were 3D printed (Objet260, Stratasys. Eden Prairie, USA) in FullCure 720 using the PolyJet process in high resolution mode (16 μm layer height). The depth of the sample area was designed to ensure the silicon surface remained at the same height as the circular magnets central axis.

The standard phosphorus-doped (n-type) c-Si 〈100〉 was utilised for ablation with an external magnetic B field, usually applied using a 3D printed plastic jig (housing 5-mm-diameter Nd${}_{2}$Fe${}_{14}$B magnets, up to four each side) with a B-field applied parallel to Si-samples surface. Also, neodymium magnets with different diameters and thicknesses were used for application of B-field perpendicular to Si-sample’s surface. The B-field strength was directly measured and ranged from $B_{ext}$ = 0.02 to 0.16 T. The conductivity of n-Si was measured by the van der Pauw method with Ecopia HMS-3000. It confirmed the n-type conductivity of $\sigma =3.76\times 10^{-4}\;\Omega^{-1}$/cm (resistivity of $\rho \equiv 1/\sigma =2.658\times 10^{3}\;\Omega$cm), bulk concentration of carriers $n_{e}=2.546\times 10^{13}$ cm${}^{-3}$, mobility of $\mu_{e}=92.25$ cm${}^{2}$/V.s and sheet resistance of $5.31\times 10^{4}\;\Omega /□$. There was no difference observed for the tested low conductivity intrinsic and n-/p-type Si in terms of formation of the ablation pattern.

All samples were cleaned in acetone/IPA prior to use to minimise surface contamination, prior to fabrication by direct laser writing. Ablation conditions varied from single pulses to heavily overlapping pulses, of low and high energy (1–1000 nJ pulses), with objective lenses of numerical aperture NA=0.14 and 0.7 (Mitutoyo). Pulse energy was measured at the entrance of an objective lens together with measured transmission coefficient *T*. Laser ablation was carried out at irradiance well exceeding the ablation threshold [31].

Strength of magnetic fields utilised was measured with a Gauss probe with 2 axis Magnetic Field sensor (Xplorer GLX PS-2002, Pasco. Roseville, USA) with several aligned magnets like that seen in the utilised sample holders. A scanning electron microscope (SEM) was used to characterise surfaces of laser ablated regions (Raith EBL 150${}^{\mathrm{TWO}}$, Raith. Dortmund, Germany).

## 5. Conclusions and Outlook

It was demonstrated that by externally applying a magnetic B-field of B∼ 0.1–0.2 T across a 5–2 mm gap with a Si sample placed inside it, the field of ablation debris and surface morphology of ablation is strongly affected. The smaller the gap (stronger B-field), the larger the periods between the micro-ablation pits observed were along the $N\overset{}{\to }S$ direction of the externally applied magnetic fields. The SPP mechanism can explain the observed pattern formation. The period of this linear pattern was not dependent on the polarisation of the incident light and was completely guided by B-field orientations. Further examination of the SPP mechanism will be necessary with focal spots favouring formation of perpendicular gradients of electron concentration and temperature, leading to spontaneous generation of magnetic fields. Future studies will also focus on observing B-field effects on the spatial sorting of high-refractive index nanoparticles.

Externally applied electric and magnetic fields can provide vectorial control of light matter interaction via the Lorentz force of photo-excited/generated carriers (electrons, holes, and ions) and enrich the available control toolbox of light–matter interactions. Engineering of permittivity via optically induced/controlled dielectric-to-metal (Die-Met) [32] transition with ultra-short laser pulses is becoming new method for re-structuring materials.

## Figures and Tables

**Figure 1 nanomaterials-09-01662-f001:**
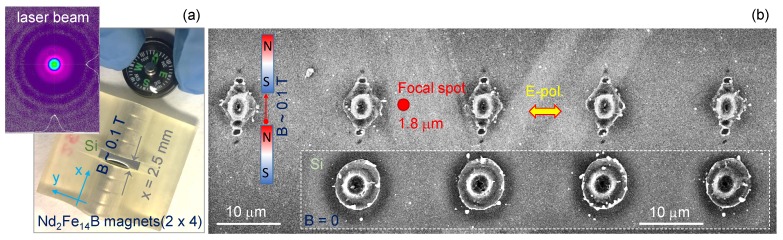
(**a**) 3D printed jig with a sample compartment gap of x=2.5 mm surrounded by Nd-magnets (four on each side to maximise B-field strength ∼0.1–0.2 T, depending on separation *x*). A (100) orientation silicon wafer was diced along the (110) plane to the required width; the magnetic field was oriented along $\langle \bar{1}10\rangle$ direction (perpendicular to the wafer base cut 〈110〉). The inset shows a Gaussian beam profile of the femtosecond laser used. (**b**) Scanning electron microscopy (SEM) images of single-pulse ablated regions with and without a magnetic field present. Laser pulses of λ=1030 nm wavelength and $t_{p}=230$ fs pulse duration were focused with an objective lens of numerical aperture NA=0.7 to a focal spot of 2r=1.22λ/NA=1.8μm diameter (comparable to the cross section at $1/e^{2}$ of a Gaussian beam). Pulse energy was $E_{p}=1.13\;\mu$J at the entrance of the objective lens (transmission of the lens was T=0.22); fluence $F_{p}\equiv E_{p}/A_{p}=9.8$ J/cm${}^{2}$; irradiance $I_{p}\equiv F_{p}/t_{p}=42.7$ TW/cm${}^{2}$ with area $A_{p}=\pi r^{2}$. For the tilted-SEM view see Figure A1.

**Figure 2 nanomaterials-09-01662-f002:**
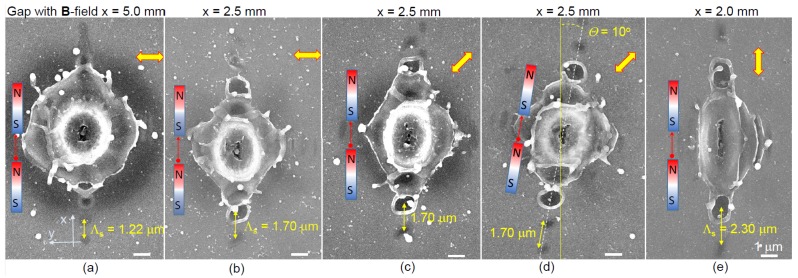
Summary of separate parameter studies: (all scale bars = 1μm) (1) when the gap between magnets is x>5 mm, seen in (**a**), subsequent ablation patterns are barely observed, similarly to having no magnetic field B=0 (Figure 1b, Figure 4a, and Figure A1); (2) polarisation rotation changes pattern only slightly ((**b**) versus (**c**)); (3) rotation of magnetic field clearly has the strongest effect of orientation of SPP pits ((**c**) versus (**d**)); (4) ((**a**) to (**e**)) period $\Lambda_{s}$ of SPP pattern depends on the gap *x* (magnetic field strength). Pulse energy and focusing conditions were like those seen in Figure 1.

**Figure 3 nanomaterials-09-01662-f003:**
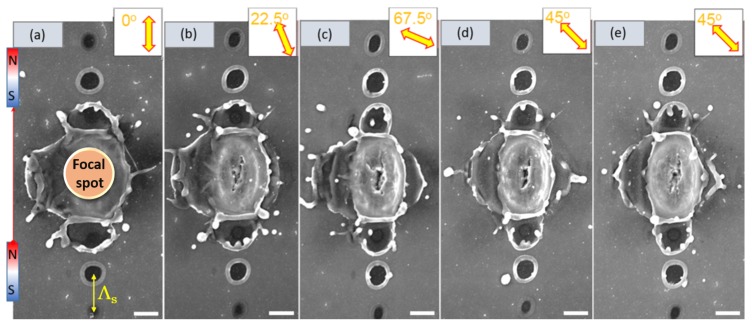
SEM images of the ablation sites under high intensity single pulse exposure on Si, with an externally applied magnetic field: (all scale bars = 1μm) all parameters and geometry were constant except incident light polarisation. The angular polarisation (E-field) dependence is shown in insets. (**a**) $0^{\circ }$ linear polarisation of incident light (parallel to B field direction) (**b**) $22.5^{\circ }$ incident light polarisation (**c**) $67.5^{\circ }$ incident light polarisation (**d**) $45^{\circ }$ incident light polarisation (**e**) $45^{\circ }$ incident light polarisation repeated. The gap between magnets x=2.5 mm and magnetic field B=0.156 T. The period $\Lambda_{s}$ of SPP pattern was constant. Pulse energy and focusing conditions were like those seen in Figure 1. For the tilted-SEM view, see Figure A1.

**Figure 4 nanomaterials-09-01662-f004:**
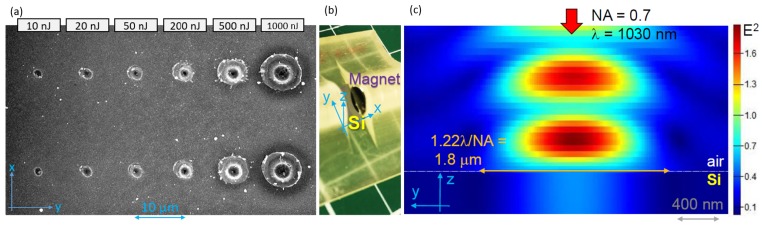
(**a**) Ablation of Si at different pulse energies where no magnetic field is applied B=0 with polarisation of light along y-axis, (at the entrance of the objective lens of numerical aperture NA=0.7; transmission of the lens was T=0.22). For the tilted-SEM view see Supplementary Figure A1 and Figure A2. (**b**) Optical image showing the placement of Si in the central part of magnetic B-field. (**c**) Finite difference time domain (FDTD) simulation (Lumerical) of light intensity distribution at the plane of incidence at the conditions of experiments. Modelling presents a linear light intensity distribution. The amplitude of the light source $E_{y}=1$.

**Figure 5 nanomaterials-09-01662-f005:**
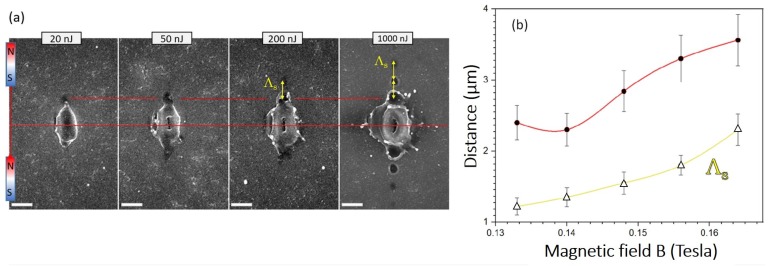
(**a**) SEM images of single pulse ablation sites at different pulse energies $E_{p}$ (at the entrance of the objective lens of numerical aperture NA=0.7; transmission of the lens was T=0.22). The gap x=2.5 mm; B=0.156 T. Scale bars = 2μm. The arrow markers are the same length of 1.75 μm, for comparisons between ablated pits formed by different pulse energies. (**b**) Period $\Lambda_{s}$ (yellow/lower curve) versus magnetic field strength (or the gap width *x*) at high pulse energy $E_{p}=1.13\;\mu$J; distance between the ablation centre and first micro-ablated pit (red/top curve) versus *B*. Lines are eye-guides; error bars are 10%.

**Figure 6 nanomaterials-09-01662-f006:**
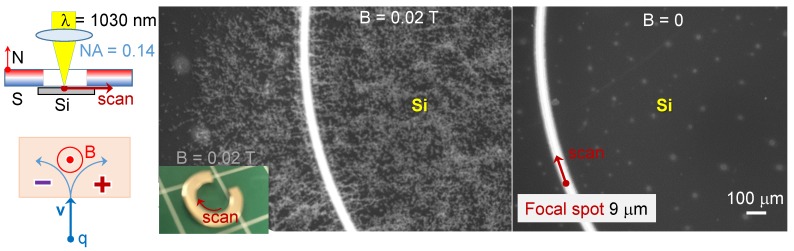
Debris formation for ablation of Si sample with a B=0.02 T magnet placed on it. Laser was scanned along the internal rim of the magnet at constant distance. For the B=0 case, the same size and shape non-magnetic material was used. Conditions: scan speed $v_{s}=100\;\mu$m/s, constant density mode with 250 pulses/μm, pulse energy $E_{p}=25$ nJ (T=0.82 transmission of the objective lens; NA=0.14), and a laser repetition rate f=200 kHz. Experimental geometry and charge separation due to Lorentz force are schematically shown.

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
