# Peer review of "External Field-Controlled Ablation: Magnetic Field"

_nanomaterials, 2019, doi:10.3390/nano9121662_

Round 1
Reviewer 1 Report
Work was improved and can be published in the present state. It containes careful studies and can be used by other researches in the field.
Author Response
Reviewer 1
Remark/Question 1. Work was improved and can be published in the present state. It contains careful studies and can be used by other researches in the field.
Answer 1. Thank you very much.
Reviewer 2 Report
While this new manuscript version has been improved compared to the previous submission, it still has numerous issues. The introduction has been expanded to place the current work in context, but I feel that the addition is too dismissive of similar literature and still does not do a good job of distinguishing the research presented here. For example, characterizing the focusing condition described in Ref. 22 as ‘weak’ on Line 26 is inaccurate. It is true that that focus was not as tight as in this work, but 11 µm features cannot be made with weak focusing. Lines 28–9 read, ‘Both these recent studies using externally applied fields fabricate with laser pulses in the nanosecond regime,’ which is grammatically awkward and untrue, as previously on Line 27 0.3 ps (300 fs) pulses were acknowledged as being employed in Ref. 22.
Furthermore, on Line 33 the newly-given motivation of spatially sorting debris nanoparticles is stated, but no such sorting analysis is presented. Why claim a major motivation when it has not been investigated? Regarding this subject, is the ablated volume the same in both cases (i.e., does the ablated trench have the same width and depth)? If so, to where does all the debris disappear in the non-magnetic case in Fig. 6? If not, and more debris is generated when the magnetic field is present, what causes the difference? Answering such motivation demands follow-through, which is lacking here.
There are six instances of the code ‘blue’ not being implemented; how much text was supposed to be colored in each case? Line 24: laser ablation is not a solid-state plasma. Line 28: I do not understand what is meant by the phrase, ‘the ablation volume was smaller’ regarding Ref. 22; the ablated volume (depth) was not reported, so how can such a conclusion be drawn? Line 81, end: avoid use of ‘you.’ Caption of Fig. 1, 8th line: ‘comparable to the 1/e2 intensity cross section…’. Lines 122–30: how is laser-assisted SiO2 formation distinguished from ablating SiO2 already natively on the surface? Silicon exposed to atmospheric oxygen will naturally form a surface oxide layer, so how can one tell that SiO2 debris was not simply from ablating pre-existing surface oxide? Was NEXFAS performed on a pristine sample for comparison? Line 145: reference should be to Fig. 4(c). Lines 146–7: sentence fragment. Line 210: a unit symbol is replaced by a square box.
Breakdown in air (stated on Line 98) implies that the focus is not ideally located at the sample surface; so, if the purpose here is ablation, why is the beam focused above the surface and not at the surface? Also, in Fig. 4(c), why are there several disconnected regions of high intensity in the air well above the Si surface? Even if the focus is displaced a small amount above the surface, shouldn’t there be only one focal spot? Lines 144–7 pertain, but what the text discusses is not clear in 4(c).
In Fig. 5(b), two different data sets are plotted, but the ordinate (vertical axis) label pertains only to the lower plot. Add a right-side label for the upper plot. Also, no discussion of the data in the upper plot is given. What is the significance of the dip in the upper plot trend at 0.14 T?
Supplement: what is the purpose of cleaving the samples? No discussion is given on this topic. Hole profiling? However, the cleavage does not intersect with holes in Figs. A1 and A2.
Figs. 3(a) and 5(a) have text annotations that are too small, and the contrast of text coloration in most figures should be reviewed, especially light grays and blues with brighter backgrounds and yellow on white (A1 and A2).
Author Response
Reviewer 2
Thank you very much for your review. Our answers follow.
Remark/Comment 1
.. characterizing the focusing condition described in Ref. 22 as ‘weak’ on Line 26 is inaccurate. It is true that that focus was not as tight as in this work, but 11 µm features cannot be made with weak focusing. Lines 28–9 read, ‘Both these recent studies using externally applied fields fabricate with laser pulses in the nanosecond regime,’ which is grammatically awkward and untrue, as previously on Line 27 0.3 ps (300 fs) pulses were acknowledged as being employed in Ref. 22.
Answer 1. Focusing is defined by the numerical aperture, NA. It is called tight when NA > 0.5 at least. Another way to define focusing as tight is when longitudinal E-field component becomes approximately 10% of the total field. This corresponds to NA > 0.7. In those cases, the focal spot is approximately two wavelengths in diameter.
Remark/Comment 2
Furthermore, on Line 33 the newly-given motivation of spatially sorting debris nanoparticles is stated, but no such sorting analysis is presented. Why claim a major motivation when it has not been investigated? Regarding this subject, is the ablated volume the same in both cases (i.e., does the ablated trench have the same width and depth)? If so, to where does all the debris disappear in the non-magnetic case in Fig. 6? If not, and more debris is generated when the magnetic field is present, what causes the difference? Answering such motivation demands follow-through, which is lacking here.
Answer 2. This analysis of spatially sorting debris is currently being done and deserves a work of its own, is not extensively presented here. As it stands, it is motivation. More debris is generated as charged material during ablation in an external magnetic field is subjected to the Lorentz force.
The non-magnetic case in fig 6 is a control, and clearly shows an disparity due to the effect of a magnetic field. For B=0, little to no debris was observed while for B>0 debris/deposition is enhanced. More studies of debris formation were presented in a specialised ablation conference COLA-2019 and it will be reported separately.
Remark/Comment 3
There are six instances of the code ‘blue’ not being implemented; how much text was supposed to be colored in each case? Line 24: laser ablation is not a solid-state plasma. Line 28: I do not understand what is meant by the phrase, ‘the ablation volume was smaller’ regarding Ref. 22; the ablated volume (depth) was not reported, so how can such a conclusion be drawn? Line 81, end: avoid use of ‘you.’ Caption of Fig. 1, 8th line: ‘comparable to the 1/e2 intensity cross section…’.
Answer 3. In most cases you mention, changes were small. One or two words, grammatical changes etc. Illustration of the ablated volume is shown in supplement. Text is rephrased.
Remark/Comment 4
Lines 122–30: how is laser-assisted SiO2 formation distinguished from ablating SiO2 already natively on the surface? Silicon exposed to atmospheric oxygen will naturally form a surface oxide layer, so how can one tell that SiO2 debris was not simply from ablating pre-existing surface oxide? Was NEXFAS performed on a pristine sample for comparison? Line 145: reference should be to Fig. 4(c).
Answer 4. Yes NEXAFS was also performed on a pristine sample of Si as reference (where small amounts of oxides are measured). There is also a large difference in oxides produced between ablation carried out with and without magnetic field.
Remark/Comment 5
Lines 146–7: sentence fragment. Line 210: a unit symbol is replaced by a square box.
Answer 5. This is correct and refers to the sheet resistance of a material, an important property especially for our discussion of phenomena presented. https://en.wikipedia.org/wiki/Sheet_resistance#Units
Remark/Comment 6
Breakdown in air (stated on Line 98) implies that the focus is not ideally located at the sample surface; so, if the purpose here is ablation, why is the beam focused above the surface and not at the surface?
Answer 6. Breakdown of air refers to photo-ionisation before the surface. Even if the focus is aimed on the surface, with our tight focusing and high irradiance condition this is still a possibility. Our two appendix figures show the same phenomena at low and high pulse energy. Hence the magnetic field is interacting with plasma in the irradiated material, not air above the surface that is ionised. Explanation is added.
Remark/Comment 7
Also, in Fig. 4(c), why are there several disconnected regions of high intensity in the air well above the Si surface? Even if the focus is displaced a small amount above the surface, shouldn’t there be only one focal spot? Lines 144–7 pertain, but what the text discusses is not clear in 4(c).
Answer 7. Please see answer above. Surface wave is launched at the Si-air interface from the strongly excited plasma (or solid-state plasma) region at the irradiation site.
Remark/Comment 8
In Fig. 5(b), two different data sets are plotted, but the ordinate (vertical axis) label pertains only to the lower plot. Add a right-side label for the upper plot. Also, no discussion of the data in the upper plot is given. What is the significance of the dip in the upper plot trend at 0.14 T?
Answer 8. The vertical axis label pertains to both sets of data, we are making a distance measurement in microns for both. The upper plot is a geometrical measurement of the first micro-ablation pit site observed from the centre of the focal spot. Discussion is added.
Remark/Comment 9
Supplement: what is the purpose of cleaving the samples? No discussion is given on this topic. Hole profiling? However, the cleavage does not intersect with holes in Figs. A1 and A2.
Figs. 3(a) and 5(a) have text annotations that are too small, and the contrast of text coloration in most figures should be reviewed, especially light grays and blues with brighter backgrounds and yellow on white (A1 and A2).
Answer 9. Cleaving samples was done to reveal the interaction volume. We also plan to carry out HF etching for hole profiling.
Thank you for the comments on figure colouring, this will be adjusted.
Round 2
Reviewer 2 Report
Answer 2: I contend that the current manuscript clearly states that nanoparticle sorting is a major motivation which is, however, not addressed here, and this is a sticking point (lines 34–5: ‘One particular motivation of this study was to observe a B-field effect on debris formation and spatial sorting of high-refractive index nanoparticles’). Giving a ‘particular’ motivation for a paper not only implies but demands that this motivation topic will be addressed in that paper. The quoted statement needs to be amended, or perhaps more appropriately moved to the conclusion as part of the Outlook/future work discussion, as it has no proper place in the introduction or body of this manuscript.
Answer 8: In Fig. 5(b), the vertical axis label refers to a specific quantity Λs for the yellow plot, defined in the text on lines 81–2 (the ‘period Λs of linearly formed pattern seen in produced micro-pits’). This quantity is not the same as the red plot’s ‘distance between the ablation center and first micro-ablated pit’ which is not a period, as there is only one ablation center and first pit. A period requires repetition. Therefore, it is incorrect to apply this labeling to both data plots. If you must use only one label, then generalize to period ‘Λ’ by removing the subscript, but annotations should be added to the plots for clarity.
Answer 9: It is still not clear what, if anything, was gained by cleaving samples. The cleaved edges do not intersect pits and expose profiles in the images shown (A1 and A2), and the sample can be tilted to 45° even if not cleaved. Moreover, how will (proposed) HF etching help in profiling the holes? Won’t etching alter the profile and defeat the purpose of profiling the existing pit?
Author Response
Thank you for the final few and benevolent remarks.
Answer 2: I contend that the current manuscript clearly states that nanoparticle sorting is a major motivation which is, however, not addressed here, and this is a sticking point (lines 34–5: ‘One particular motivation of this study was to observe a B-field effect on debris formation and spatial sorting of high-refractive index nanoparticles’). Giving a ‘particular’ motivation for a paper not only implies but demands that this motivation topic will be addressed in that paper. The quoted statement needs to be amended, or perhaps more appropriately moved to the conclusion as part of the Outlook/future work discussion, as it has no proper place in the introduction or body of this manuscript.
ANSWER. Rephrased.
Answer 8: In Fig. 5(b), the vertical axis label refers to a specific quantity Λs for the yellow plot, defined in the text on lines 81–2 (the ‘period Λs of linearly formed pattern seen in produced micro-pits’). This quantity is not the same as the red plot’s ‘distance between the ablation center and first micro-ablated pit’ which is not a period, as there is only one ablation center and first pit. A period requires repetition. Therefore, it is incorrect to apply this labeling to both data plots. If you must use only one label, then generalize to period ‘Λ’ by removing the subscript, but annotations should be added to the plots for clarity.
ANSWER. Thank you. Figure is updated to more clearly presents different distances measured from SEM images.
Answer 9: It is still not clear what, if anything, was gained by cleaving samples. The cleaved edges do not intersect pits and expose profiles in the images shown (A1 and A2), and the sample can be tilted to 45° even if not cleaved. Moreover, how will (proposed) HF etching help in profiling the holes? Won’t etching alter the profile and defeat the purpose of profiling the existing pit?
ANSWER. Rephrased. The side view shows the ablation profile more clearly since the contrast in SEM imaging is due to charging and not necessary due to topography. The slanted view clearly reveals the ablation volume.
This manuscript is a resubmission of an earlier submission. The following is a list of the peer review reports and author responses from that submission.
Round 1
Reviewer 1 Report
Work is interesting, well illustrated and requires only a minor revision.
Introduction is very short and it contains massive citations which are given without proper explanation/description of the points of interest. Introduction must include detailed state of the art description and clear goal of the present study. In the present state it is a diffused description of the experiment mixed with experimental details.
Figure 1 requires coordinates schematic description. Authors could also discuss why 0.1 T field was selected.
There are a number of misprints and bad formatting symbols (5.31 × 10 181
4 Ω/..., etc.) which must be corrected.
Reviewer 2 Report
In this paper the authors show that magnetic field of relatively low amplitude of 0.1 T can be used to control formation of ablation patterns and debris during irradiation of silicon by femtosecond laser pulses. The results are based on observation of periodic spots around the main ablated spot, whose properties depend on the presence and direction of the magnetic field but not on the laser polarization. The origin of these structures is explained using surface plasmon polariton waves in combination with the ion and electron propagation along the field lines of the magnetic field.
The experimental part of the paper is clearly written and the presented results are definitely interesting. However, in the part discussion, there are several points, which are not completely clear to me. I have several questions and comments to the paper, which should be addressed prior to its acceptance.
1) In the part describing surface plasmon polariton, the authors present Eqs. (1) and (2) and they say, that these equations describe “phase-matching conditions for the SPP wave vector”. I don´t understand this part. I assume that phase-matching between the incident light and SPP is required for efficient SPP excitation. Eqs. (1) and (2) only describe the SPP wavevector, not the phase-matching condition. Please clarify this part.
2) In a tightly focused laser beam, not only the transverse fields but also the longitudinal component of the field in the direction of light propagation can play role. Can the authors estimate the amplitude of the longitudinal field component? Could it play some role in SPP excitation?
3) Is it possible to estimate how long time the hot plasma (ions and electrons) needs to get back to the surface and form the periodic patterns? Does the timescale match the decay time of SPP on silicon-air interface?
Minor points:
Which definition is taken for the laser spot size? 1/e^2 radius, FWHM or other? Please add this information somewhere in the manuscript.
Typos:
Line 66: “The period of ablated pits observed DEPENDS ....”
Line 127: “... which IS proportional ...”
Reviewer 3 Report
This manuscript describes using an externally applied static magnetic field to control the ablation dynamics of silicon by femtosecond laser pulses. However, this topic has already been addressed in the literature: H. Tang et al., “Repulsive magnetic field-assisted laser-induced plasma micromachining for high-quality microfabrication,” Int. J. Adv. Manuf. Tech. 102, 2223–9 (June 2019), and similarly, V. Gruzdev et al., “Fundamental mechanisms of nanosecond-laser-ablation enhancement by an axial magnetic field,” J. Opt. Soc. Am. B 36, 1091–1100 (April 2019), neither of which have been cited here. In light of these published articles, there is nothing in the present manuscript that distinguishes this work. Moreover, machining applications normally seek to reduce debris production and contamination, but the results here show the opposite effect when applying a magnetic field (e.g., Fig. 6). It is therefore difficult to understand how this technique assists the fabrication process, and the authors have not discussed the significance of this (undesirable?) outcome.
The introduction is severely lacking. It provides a short list of applications for ultrashort laser material modification (with some references poorly/incorrectly cited) but does not establish any context for why applying an external magnetic field is important or useful, nor does it explore the history of applying a magnetic field during laser machining.
The figures and captions are, in general, not well-constructed, with too much (and often repetitive) information, low-contrast overlay coloring, overlay measurement information that conflicts with the caption, and inconsistent sizing of image panels that yields a sloppy appearance.
The References contain numerous errors, to a level that should be considered inappropriate for a submission. More than half have at least one problem: missing/incorrect initials, names, or title words; incorrectly sloped accents or missing accents in authors’ names (consistency, please—if some names can have accents, all accents should be applied properly); incomplete pagination; incorrect/missing volume/article number. Some journal abbreviations are inconsistent.